behaviour

aggression, intra-sexual selection, ontogeny, social dominance, social rank, social status

**Author for correspondence:**
Olof Leimar
e-mail: olof.leimar@zoologi.su.se

# Effects of social experience, aggressiveness and comb size on contest success in male domestic fowl

Anna Favati[1], Hanne Løvlie[2] and Olof Leimar[1]

[1]Department of Zoology, Stockholm University, 10691 Stockholm, Sweden
[2]Department of Physics, Chemistry and Biology, IFM Biology, Linköping University, 58183 Linköping, Sweden

HL, 0000-0003-4352-6275; OL, 0000-0001-8621-6977

The ability to dominate conspecifics and thereby gain access to resources depends on a number of traits and skills. Experience of dominance relationships during development is a potential source of learning such skills. We here study the importance of social experience, aggressiveness and morphological traits for competitiveness in social interactions (contest success) in male domestic fowl (*Gallus gallus domesticus*). We let males grow up either as a single (dominant) male or as an intermediately ranked male in a group of males, and measured their success in duels against different opponents. We found that single-raised males had lower contest success than group-raised males, and that aggression and comb size correlated positively with contest success. This indicates that experience of dominance interactions with other males increases future success in duels. We similarly studied the consequences of growing up as a dominant or subordinate in a pair of males, finding no statistically significant effect of the dominance position on contest success. Finally, we found that males were consistent over time in contest success. We conclude that social experience increases contest success in male domestic fowl, but that certain behavioural and morphological characteristics have an equal or even stronger covariation with contest success.

## 1. Introduction

The factors influencing contest success, including social skills, have been thoroughly studied from a fitness perspective. For example, it is well known that dominant individuals commonly enjoy higher access to important resources such as food and mating partners, and thereby may gain higher reproductive success [1]. Less is known about the proximate causes and development of traits

influencing contest success, which in general are influenced by both heritable and environmental factors [2–4]. With the exception of species where social dominance is inherited (e.g. [5]), an individual is not in itself dominant or subordinate, but becomes so depending on the social competition it meets. Therefore, the underlying mechanisms should allow flexibility [6,7]. Indeed, dominance status, with its correlated behaviours and associated physiological mechanisms, is often plastic and may be changed experimentally [6–10]. In addition, the social environment experienced early in life can have a long-lasting influence on an individual's behavioural profile [11], affecting how the individual handles social challenges in adulthood, including intra-sexual competition [12–15] and territoriality [16,17]. For example, in golden hamsters (*Mesocricetus auratus*) social defeat during puberty causes neurochemical, immunological, hormonal and behavioural changes in young adult males [18]. Juveniles are predicted to adjust their social behaviour to the prevailing social situation, in order to optimize future costs and benefits of social interactions when becoming young adults [19,20]. A mechanism that could serve as a link between the juvenile social situation and adult behaviour is state dependence of behaviour. Individuals can vary in different slow-changing 'states' (e.g. energy reserves, size or vigour) that alter the costs and benefits of behaviour [21–23]. This means that individuals with different states could differ consistently in behaviour, at least as long as the state remains. In combination with positive feedback systems, this could result in different behavioural phenotypes [23–25], or personalities (among-individual consistency in behaviour [21,26,27]). Social conditions, including social dominance, have been proposed to constitute such states [25], but this has rarely been tested empirically. There are studies finding short-term effects of social status on behaviour (e.g. [10]), yet the long-term effects of having a certain status are unknown. If early social experiences lead to different trajectories in life, this may cause or cement differences in contest success and also influence other behavioural traits. Alternatively, early (genetically) determined personality or morphological traits may influence the dominance position an individual obtains [28], thereby modulating its social environment and experiences. Up to now, it is not known whether early social experiences or early determined behavioural traits are more influential in the development of adult contest success.

We here study if dominance interactions during the ontogeny of male domestic fowl (*Gallus gallus domesticus*) affect adult contest success (measured as the probability of winning intra-sexual duels over dominance). The domestic fowl is a group-living species that forms sex-specific dominance hierarchies. These are often linear, meaning that the top dominant individual is dominant over all individuals of the same sex; the second is dominant over all but the first, and so on, down to the most subordinate individual [29]. The hierarchy in a group is relatively stable and a male can be dominant for a whole breeding season, and sometimes even for several years [30]. Under natural conditions, groups consist of one or several males and an equal or larger number of females [30]. Dominant males are more active, crow more, and are more vigilant, compared with subordinate males [10,31]. The propensity to become dominant is partly determined by the physical qualities of the male, and can be higher for males with larger comb size (a fleshy, red ornament on the head, e.g. [32]) or body size. Behaviour, such as aggressiveness and explorative behaviour, can also correlate with dominance [33,34]. Aggressiveness has been found to be positively related to reproductive success, as estimated using molecular data on red junglefowl (*Gallus gallus*) [35].

It is unknown whether social experiences during development from chick to adult affects the propensity to win fights over dominance. It has been observed in both birds and mammals that males reared in male–female pairs during young ages can become more aggressive compared with group-reared males [15,36]. Anecdotal evidence similarly suggests that male domestic fowl that are kept singly in female groups become very aggressive and difficult to house with other males. On the other hand, social experience is expected to increase social skills, including adequate submissive behaviours [17]. By letting males grow up as either the single dominant male or as a male of intermediate rank in a group of males, we investigate if future contest success in pairwise social interactions is influenced by social experience. We also investigate if aggressiveness, body size and comb size differs between single- and group-reared males and if these traits covary with contest success. Further, we examine if aggression covaries with other personality traits, measured in a novel arena (NA) assay. In a similar way as for the single- and group-reared males, we let males grow up in pairs, with one becoming dominant and the other subordinate, and study the subsequent effect on contest success.

## 2. Material and methods

Several of our methods were similar to those that have been used previously. Among these were the determination of dominance relations [10,34] and the scoring of aggression and exposure to an NA [33,34,37–39].

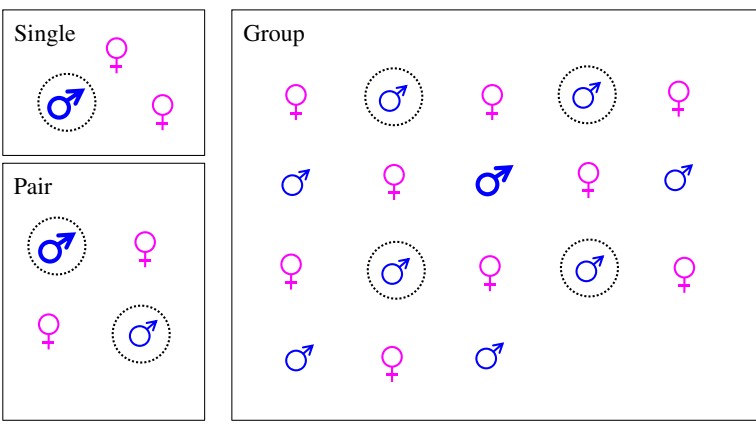

**Figure 1.** Experimental set-up for the social treatments of focal males. Single, Group and Pair are the social treatments, for testing effects of social rearing on contest success in male domestic fowl. Each box in the figure represents a replicate, containing males and females from the experimental rearings. The bolder male symbols represent the dominant male in the replicate and the dotted circles indicate focal males. The focal males underwent a number of tests, in the form of duels against opponent males from the main population, aggression assays and NA assays (figure 2).

## 2.1. Animals and housing

The study was conducted during 2013–2015 at Tovetorp research station, Stockholm University, Sweden, using domestic fowl of the breed 'Swedish bantam' (*G. g. domesticus*, 'Gammalsvensk dvärghöna'). Birds of this population appear and behave similarly to that of the ancestral red junglefowl (*Gallus gallus*) [40], and were kept under semi-natural conditions (population size 2013 = 106, 2014 = 118). Randomly selected eggs from this population were artificially incubated in two batches (hatched in June 2013 and June 2014) producing birds for the social treatments, including the focal males used in the experiment. Chicks were raised in a common garden set-up, and sexed by visual inspection of secondary sexual traits (feather colour and comb size) that first appears at around two to three weeks of age. During the experiment, birds were housed in indoor aviaries (sized approx. 2 m$^2$ for the Single treatment, approx. 6 m$^2$ for the Group treatment and approx. 2.5 m$^2$ for the Pair treatment; see below), with access to a roofed outdoor area (approx. 6 m$^2$) every second to fourth day on a balanced, rotating schedule. Aviaries were visually, but not auditorily separated. All birds had ad libitum access to food and water, and aviaries had wood shavings on the floor, were fitted with a perch, dust bath and nest-box, and were regularly enriched with fresh branches, tallow balls and sprinkled whole grain. The aviaries contained a heat roof until birds were 10 weeks old.

## 2.2. Social treatments

Males from the experimental rearings were randomly assigned to grow up in aviaries under one of three different social conditions (figure 1), as follows: 'Single', a single (dominant) male, together with two females (four males in 2013 and four in 2014); 'Group', a group of males and females (14 males in the group in 2013 and nine in the group in 2014); and 'Pair', a pair of males, each together with two females (three pairs in 2013, one male in a pair died before testing in 2013, and four pairs in 2014). Females from the experimental rearings, hatched at the same time point, were randomly assigned as company in the treatments. From these social treatments, we selected focal males for two different comparisons: first, a comparison between 'Single' (eight males) and 'Group' (eight males), where group-reared focal males were middle ranked in their groups (see below) and, second, a comparison between dominant and subordinate males of pair-reared males.

Dominance relationships among males within the Group and Pair rearings were determined using direct observation of agonistic behaviour. Three to five avoidances in a row from the same male were used to assign a male as subordinate to the other male, who was considered dominant. In the Group treatment, the two males of the highest social rank and the two of lowest rank were determined using this method, and focal males were randomly chosen among the remaining middle-ranking males. Our aim was that these focal males should experience dominance interactions during ontogeny, including both wins and losses. For the Pair males, dominants and subordinates might differ both in traits already determined at pair formation, such as aggressiveness, and in their experiences in the pair, so

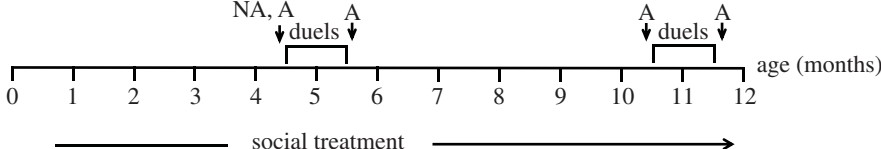

**Figure 2.** Time line of social treatments, duels and behavioural assays when testing effects of social rearing on contest success in male domestic fowl. The social treatments (figure 1) started when focal males were three weeks old. Two rounds of duels between focal and opponent males, with 6–7 duels per focal male, were performed six months apart. Aggression was scored one week before and one week after each duel round (A), for both focal and opponent males. Two weeks before the first duel round, an NA assay was performed for the focal males. Exposure to NA was repeated one week later.

our dominant–subordinate comparison may involve a combination of different factors. This is the reason why we analysed this comparison separately. As is generally the case for domestic fowl, in all instances males were observed to be dominant over females in the social treatments.

At three weeks of age and until 1 year of age, focal males underwent a series of behavioural tests, in the form of duels, aggression and NA assays (figure 2). Dominance rank of focal males was assessed regularly during the course of the experiment, which lasted until the birds were around 1 year old. One pair of males in the Pair treatment changed dominance relationship in the winter between duel periods (figure 2). There were around four months between the change in status and the following duel period, which is the same amount of time as between hatching and the first duel period, and therefore the current social status of these two males at the time of each duel period was used in statistical analyses of focal Pair males.

## 2.3. Duels

To investigate if the social experience influences contest success, two rounds of staged duels were performed: one starting when focal males were 4.5 months old, thus when they had recently become sexually mature [37], and one starting six months later when they were 10.5 months old (figure 2). We implemented a design where a focal male met six (in the first round of duels) and seven (in the second round of duels) unfamiliar randomly chosen conspecific opponent males from our main population (age 3–5 years). We refer to these males from the main population as opponent males. Subordinate focal males from the Pair treatments did, however, not participate in the first round of duels. Each focal male met the same opponent males in random order. Opponent males were kept in a separate flock together with females (sex ratio approx. 1 : 1) between duels. The opponent males were not the same for the two rounds of duels, resulting in a total of 13 duels with unique opponent males for a focal male. Opponent males had 2 days of rest between each duel, and the focal males had 4 days of rest between each duel. Duels were conducted in a room (2 × 3 m) with wood shavings on the floor and a water container in one corner. Duels were directly observed from outside this room, through a small window and video filmed using a Go Pro 3 camera. Duels lasted for up to 10 min and were prolonged for another 10 min if a clear winner had not yet been determined. If no winner was determined after 20 min, the duel was considered to end in a draw, and the data were excluded from the analyses of wins and losses (this happened in 28 out of 330 duels). Most duels were settled very quickly, with no physical contact and only aggressive displays. When a male avoided the other male at least three times in a row, he was considered the loser, and the other the winner. Reversed avoidances were rare, and if they occurred, the pair was observed until three avoidances in a row from the same male were observed. If one male was behaving fiercely towards the other male despite the other signalling submission, or if a male was injured, the duel was interrupted and the two males separated. This only happened on rare occasions.

Three males were injured by a hit to the eye during the first round of duels. One male (from the Pair treatment) did not recover and was later euthanized. Data from the two duels he participated in are included in the analyses. The other two males (from the Single treatment) were paused from the duels until they had fully recovered and showed no signs of injury, and could then continue the duel series.

## 2.4. Personality assays

To investigate if aggressiveness differed between social treatments and/or was related to contest success, we performed aggression assays (for both focal and opponent males). Aggressiveness was estimated by scoring the behavioural response of a male towards an unfamiliar intruder male ($n = 8$) that was

restrained in the hands of the observer. The male was gently herded to a corridor (approx. $0.75 \times 2$ m) between the home aviary and the outdoor enclosure, thus a familiar environment, while the rest of the flock remained in the aviary. An experimentally presented male intruder was held standing on the ground 1 m in front of the male. Aggressiveness was scored on a scale from 0 to 6, where 0 was the least aggressive response (avoiding) and 6 the most aggressive (immediately attacking; see [34] for the full scale). For focal males, aggressiveness was estimated one week before and one week after each duel round (figure 2), resulting in four scoring occasions in total. Aggressiveness of the opponent males was estimated once close in time to each duel round.

Further, to investigate if aggressiveness was related to other personality traits, focal males were exposed to an NA around two weeks prior to the first round of duels. The NA assay was repeated after one week. The NA assay was performed in an indoor arena ($3 \times 4$ m) with wood shavings sprinkled on the floor and beige tape on the floor dividing the arena into 25 sub-areas. Five familiar objects (first trial: plastic spruce trees, second trial: green plastic buckets) were placed on the floor to obstruct the males' view of the full arena, to encourage exploration. Males were tested singly and were placed on one side of the arena with the lights off. The observer moved out, turned on the light and observed the bird for 10 min, through a one-way window. Responses to the NA were used to estimate exploration and vigilance. To estimate exploration, a combined measurement consisting of the mean of three $z$-standardized responses was used: the number of sub-area transitions, the latency to enter five sub-areas (multiplied by −1 to achieve a variable where higher values correspond to a more explorative response), and the number of sub-areas visited. Vigilance was estimated as the proportion of time spent with the head above shoulder height, one observation every 30 s. The number of territorial crows was also noted.

## 2.5. Morphological measurements

Size of the comb (a status signalling ornament on the head) can be positively associated with social status [32,41]. Comb size (estimated by comb length, measured to the nearest tenth of a millimetre with a digital calliper) of both focal males and the duel opponents were therefore measured directly before each duel round. We also measured body weight (measured to the nearest gram with a digital scale) to investigate if the social treatment affected body weight, and to control for possible effects of body weight on contest success.

## 2.6. Statistical analyses

### 2.6.1. Factors affecting contest success

A total of 302 duels were resolved within the given time frame and were used in the analyses. To investigate the effect of aggression, comb size and body weight on contest success, a relative value of each variable was calculated by subtracting the value of the opponent from the value of the focal male. The reason for including aggression here is that it can influence the outcome of dominance interactions [33,34]. The probability of focal males winning duels against opponent males was analysed using a generalized linear mixed model with the outcome (won/lost) as a binary response variable (logistic regression). Treatment (Single and Group), or Dominant and Subordinate from the Pair treatment, relative aggression score, relative comb size, relative body weight and batch (2013 and 2014) were used as fixed effects, and the identity of the focal male was set as a random effect. All quantitative variables were $z$-standardized.

### 2.6.2. Personality and morphological traits

We analysed if personality measures estimated from the NA assay covaried with aggressiveness by using multiple regression, with focal males as data points. Exploration, vigilance and number of crows were averaged over the two NA tests and aggressiveness was averaged over four scoring occasions. Personality measures were $z$-transformed to ease comparisons of effects. For further illustration, we also tested for differences in personality and morphological traits between focal males from the Single and Group treatments, as well as for a correlation between aggressiveness and comb size among these males.

### 2.6.3. Consistency of contest success

To investigate consistency of contest success, we compared the proportion of duels won by a male in the two duel rounds (autumn and spring) using correlation and also by estimating repeatability (intra-class correlation, ICC), by using the R package rptR [42,43] with 1000 parametric bootstraps.

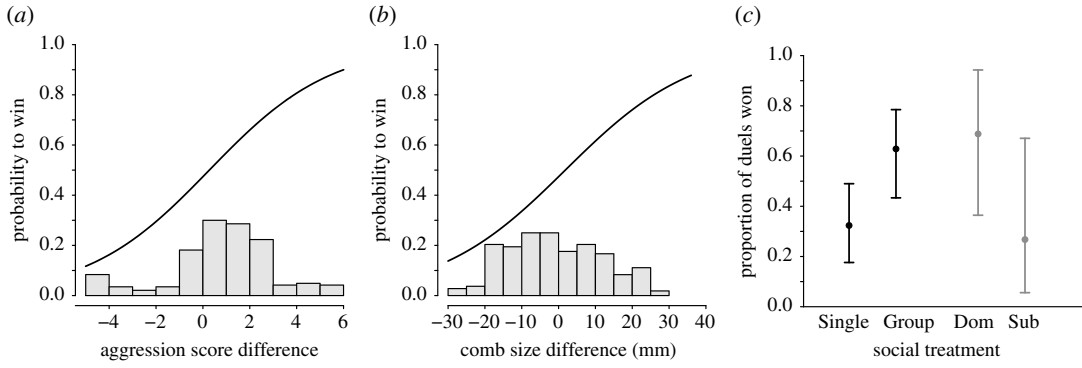

**Figure 3.** The probability of winning over two rounds of duels between focal and opponent males (see table 1 for statistical analysis). (*a*,*b*) Logistic regression curves from a fitted model like the one in table 1, illustrating the importance of aggression (*a*) and comb size (*b*), relative to the opponent male, for the probability of the focal male to win, based on focal males from the Single and Group treatments. The inserted histograms show the distributions of the focal–opponent differences in aggressiveness and comb size. (*c*) The proportion of duels won (mean ± 95% profile confidence intervals from a model like that in table 1) for focal males (*n* = 16) from the Single and Group treatments, and in lighter grey, the proportion of duels won (mean ± 95% profile confidence intervals from a model like that in table 2) for focal dominant males and focal subordinate males from the Pair treatment (*n* = 13).

**Table 1.** Statistical analysis of the outcome of duels between focal and opponent males: effects of Single versus Group social treatment and focal–opponent differences in aggression, comb size and body weight. The binary response variable is the outcome in duels against a sequence of opponent males (win/loss for a focal male; 186 duels). The focal male identity (16 males) is included as a random effect (given as the s.e.). The Single versus Group comparison is a two-level factor. Other fixed effects are the *z*-standardized differences (focal–opponent) in aggressiveness, comb size and weight, and batch as a two-level factor. Estimated effects are on the logit scale.

| effect | estimate | s.e. | P |
|---|---|---|---|
| treatment: Single versus Group | −1.26 | 0.49 | 0.010 |
| aggressiveness difference | 0.89 | 0.27 | 0.001 |
| comb size difference | 0.73 | 0.23 | 0.001 |
| weight difference | −0.27 | 0.22 | 0.21 |
| batch: 2014 versus 2013 | −1.84 | 0.50 | <0.001 |
| focal male ID | 0.55 | | |

# 3. Results

## 3.1. Factors affecting contest success

In our first comparison, between Single and Group, focal males from the Single treatment lost more duels compared with those from the Group treatment (table 1 and figure 3). The difference between the focal male and the opponent in aggressiveness and comb size, as well as batch year, also had a statistically significant influence on the chance of winning duels (table 1 and figure 3), but the effect of the difference in weight was not statistically significant (table 1). In our second comparison, dominant and subordinate pair males did not show a statistically significant difference in contest success (table 2 and figure 3*c*). We do not have an explanation for the difference between years (Batch) in tables 1 and 2, but in principle there could be differences either in focal or in opponent males (or in both) between years.

## 3.2. Personality and morphological traits

Aggressiveness did not differ significantly between focal males from the Single and Group treatments ($F = 1.39$, d.f. = 1,14, $p = 0.26$) and was not significantly correlated to comb size for these males (Spearman $r = −0.18$, $p = 0.51$, 16 males). Exploration and crowing also did not differ significantly

**Table 2.** Statistical analysis of the outcome of duels between focal and opponent males: effects of Subordinate versus Dominant social status of focal Pair males and focal–opponent differences in aggression, comb size and body weight. The binary response variable is the outcome in duels of focal males against a sequence of opponent males (win/loss for a focal male; 116 duels). The focal male identity (13 males) is included as a random effect (given as the s.e.). The Subordinate versus Dominant comparison is a two-level factor. Other fixed effects are the $z$-standardized differences (focal–opponent) in aggressiveness, comb size and weight, and batch as a two-level factor. Estimated effects are on the logit scale.

| effect | estimate | s.e. | $p$ |
|---|---|---|---|
| Subordinate versus Dominant | −1.80 | 1.10 | 0.10 |
| aggressiveness difference | 1.16 | 0.63 | 0.07 |
| comb size difference | 0.61 | 0.42 | 0.14 |
| weight difference | −0.28 | 0.47 | 0.56 |
| batch: 2014 versus 2013 | −4.51 | 1.20 | <0.001 |
| focal male ID | 1.09 | | |

**Table 3.** Statistical analysis of the effects of personality scores from NA assays on mean measured aggressiveness. The estimates are the coefficients in a multiple regression (for 29 focal males) of $z$-standardized mean personality scores on mean aggressiveness.

| effect | estimate | s.e. | $p$ |
|---|---|---|---|
| exploration | 0.28 | 0.23 | 0.22 |
| vigilance | 0.03 | 0.20 | 0.89 |
| crowing | −0.27 | 0.22 | 0.22 |

between focal males from the Single and Group treatments, but focal Group males were less vigilant compared with Single males ($F = 5.53$, d.f. $= 1,14$, $p = 0.03$). There were no significant differences between focal males from the Single and Group treatments in either body weight (mean ± s.e., Single: 1047 ± 26 g, Group: 1089 ± 26 g, $F = 1.27$, d.f. $= 1,14$, $p = 0.28$) or comb size (mean ± s.e., Single: 75.5 ± 2.5 mm, Group: 73.1 ± 2.5 mm, $F = 0.45$, d.f. $= 1,14$, $p = 0.51$). None of the personality measures from the NA assay (exploration, vigilance, crowing) had a statistically significant covariation with aggression (table 3).

## 3.3. Consistency of contest success

Males were highly consistent between the two duel rounds (autumn and spring) in their contest success (21 males; Spearman $r = 0.75$, ICC $= 0.61$, 95% CI: 0.27, 0.82, figure 4).

## 4. Discussion

Our study examined the combined importance of social experience during ontogeny, aggressiveness and morphological traits on contest success in male domestic fowl. In our first comparison of contest success against opponent males, between focal males from the Single and Group treatments, the social experience was experimentally imposed, allowing us to draw conclusions about causal relations. We found that social experience during rearing promotes intra-sexual contest success. Single males, who had limited social interactions with other males and behaved as dominants, were less successful in contests than intermediately ranked Group males. Thus, the social experience of Group males, consisting of winning and losing dominance interactions, promoted contest success more strongly than the Single males' own experience of being dominant. It could be that Group males in our study became more skilled in contests over dominance. This type of consequence of greater fighting skills has been discussed [44] and investigated [45] in previous work. In addition to an effect of social experience, we found that the relative aggression score and comb size of the competing males were positively correlated with contest success (table 1 and figure 3). Comparing the ranges of variation in the probability of winning (figure 3), it seems that aggression and comb size both were more

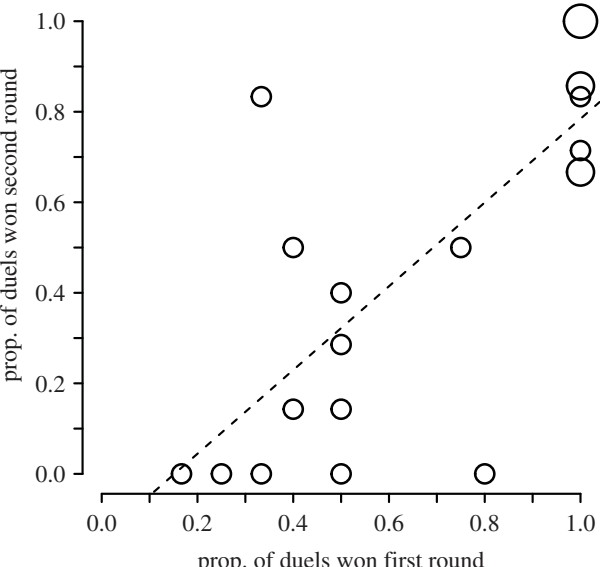

**Figure 4.** The proportion of contests won in each of two rounds of duels (autumn and spring) by focal male domestic fowl. The area of the circles is proportional to the number of males with the same proportions of contests won. The dashed line shows a linear regression ($F = 24.9$, $p < 0.001$, $n_{males} = 21$).

important than social experience for contest success. We also found that a focal male's contest success was highly consistent over time (figure 4).

In our second comparison of contest success, between dominants and subordinates of focal Pair males, our experimental design entailed that the dominance status of Pair males was not randomly assigned but rather was a consequence of the interactions in the pair. There is thus a possibility that a male's characteristics, for example his aggressiveness, influenced his dominance status in the pair, so that in principle these characteristics and the experience of a dominance position in the pair together could have influenced the contest success for focal Pair males. We found no statistically significant effect of dominance status on contest success in the comparison between dominants and subordinates (table 2 and figure 3). However, because of a small sample size and perhaps random variation in contest outcomes it is not possible to draw any firm conclusions. It thus remains to be determined whether or not winning as compared with losing experiences during ontogeny in male domestic fowl has a longer-term influence on contest success.

In our study, focal male comb size (compared with the opponent) was positively related to winning duels (table 1). Comb size was uncorrelated to aggression, which indicates that comb size correlates with some other trait that is beneficial in contests, possibly mediated by testosterone [32,46], and that comb size could act as a signal of the male's abilities or other characteristics that is not strongly correlated with his aggressive tendencies.

We found that aggression was positively correlated with contest success (table 1), but none of the behavioural responses in the NA assay covaried with aggression (table 3). Of our other personality measures, exploration is sometimes, and sometimes not, found to covary with the outcome of social interactions in male fowl [33,47], which could indicate that exploration has a rather weak effect on the establishment of social dominance, and that it only matters when contestants are closely matched for other important traits such as aggression and comb size (as a comparison, a correlational study found that social status was not associated with exploration in a cooperatively breeding warbler, *Acrocephalus sechellensis*, [48]). Closer examination of the circumstances under which exploration influences contest success is needed before firm conclusions about this relationship can be drawn.

We found that males raised in groups were less vigilant compared with males raised singly. The explanation for this could be the much studied general relationship between group size and vigilance [49]. Thus, a larger group size for Group compared with Single focal males might make the Group males accustomed to a lower need for vigilance.

An individual's social environment during ontogeny might shape individual behaviour, which could become stabilized through, for example, learning or organizational effects of hormones [50]. Long-lasting positive effects of early experiences of winning of social interactions on future contest success have been

found in genetically identical (clonal) fish (Amazon molly, *Poecilia formosa*, [20]), although it is not clear if this is related to social hierarchies in this species under field conditions. In our study, we expected focal males that were raised as a single, and therefore dominant, male would be more aggressive in interactions with other males, due to an overestimated (and untested) assessment of their own fighting capacity. Contrary to our expectations, we observed no significant differences in aggressiveness between Single and Group focal males, and even a reduced contest success for Single focal males. Our finding differs from observations on zebra finches (*Taeniopygia guttata*) [15] and guinea pigs (*Cavia aperea f. porcellus*) [36], where males reared as the single male in male–female pairs during adolescence were more aggressive when later competing for a female, compared with males reared in larger mixed-sex groups. It has been suggested [15] that single-reared males may not be properly adapted to group living, in agreement with the observation that group-reared males were better socially integrated, and that single-reared males showed less aggression with time when adjusting to a group context [51]. It has been found that social complexity or variability in the early social environment may enhance adult contest success [2,52]. Our results indicate that social experience, in the form of interactions with other males, can enhance contest success.

Earlier studies of the same population of fowl have found that when social dominance relationships remain stable for a longer period of time, the dominance relationships in the home group do not strongly influence behaviour when a male is briefly taken out of his social context and behaviourally tested with no group members present [47]. Our present results add to the impression that although social roles may determine behaviour within the group [8], or have strong short-term effects (winner–loser effects, e.g. [34]), these effects are not necessarily long-lasting once an individual is moved outside of its social context. Instead, we found correlational evidence of a different kind: the personality trait aggression could affect an individual's chances of winning agonistic social interactions [33,53–55]. The results of our study suggest that aggression develops largely independently of social experience, and that aggression is associated with the chance of winning intra-sexual contests. Such a possible importance of aggression for social dominance, in red junglefowl as well as in domestic fowl, is supported by several previous studies [56].

In conclusion, we have demonstrated that contest success in male domestic fowl is enhanced by experiences of intra-sexual competition, perhaps through increased fighting skill [44], and also shows notable covariation with individual differences in aggression and comb size.

Ethics. All applicable international, national and institutional guidelines for the care and use of research animals were followed. All procedures performed were in accordance with the ethical standards in Sweden, and the study was approved by the Linköping Ethical Committee (ethical permit number 60–10), which is the relevant national agency for ethics evaluation of research with animals at Stockholm University field station.

Data accessibility. Data and R code for the statistical analyses are available from the Dryad Digital Repository: https://doi.org/10.5061/dryad.n2z34tmtn [57].

Authors' contributions. A.F. designed the study, performed the experiments, collected data, carried out the statistical analyses and wrote the manuscript. H.L. helped draft the manuscript. O.L. conceived of the study, helped design the study, carried out data analyses and drafted the manuscript. All authors contributed to the editing of the final manuscript.

Competing interests. The authors declare that they have no conflict of interest.

Funding. A.F. was funded by Alice och Lars Siléns fond; H.L. was supported by the LiU programme 'Future research leaders' and O.L. was funded by the Swedish Research Council (grant no. 2018-03772).

Acknowledgements. We are grateful to Charlotte Rosher and Caroline Sollevi for assistance with data collection.

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
