## [Peer Review File · Royal Society Open Science]

Review History

RSOS-201213.R0 (Original submission)

Review form: Reviewer 1

Is the manuscript scientifically sound in its present form?

Yes

Are the interpretations and conclusions justified by the results?

Yes

Is the language acceptable?

Yes

Do you have any ethical concerns with this paper?

No

Have you any concerns about statistical analyses in this paper?

No

Recommendation?

Accept as is

Comments to the Author(s)

The paper is not bringing breakthrough information, however, the labour and time intensive experiments are well designed and results are presented in clear and concise way. In addition to Fig. 1, an infographics describing experimental setup would be useful. Not applicable for this manuscript, however, it would be beneficial to have information about the testosterone levels in tested males. Along with comb length, perhaps spur length and wattle size could add relevant information.

Review form: Reviewer 2**Is the manuscript scientifically sound in its present form?**

Yes

Are the interpretations and conclusions justified by the results?

No

Is the language acceptable?

Yes

Do you have any ethical concerns with this paper?

No

Have you any concerns about statistical analyses in this paper?

No

Recommendation?

Major revision is needed (please make suggestions in comments)

Comments to the Author(s)

The present manuscript reports the results of an experiment that investigated the relationship between social experience and contest outcome, and secondarily, whether social experience influences personality according to the Introduction (the Discussion takes a different tack and states that the purpose of the study was to investigate the influence of social experience and personality on fighting ability). I think some serious re-drafting of the text is required as I had some difficulty understanding what variables were supposed to be influencing what. There is also the lack of consistency throughout the paper; for example, fight outcome and individual aggressiveness (which are not the same thing) are referenced but I was under the impression that outcome was the variable of interest.

I think the main problem I had was with the somewhat circular nature of the rationale underlying the experiment. It wasn't clear to me whether social experience was expected to influence personality. From my reading of the Introduction I think this is the case; however, we need to know the personality of the individual before they are exposed to the social group element of the procedure. This is not what has happened here where social groups were set up and personality subsequently tested.

I was interested in why the authors felt that outcome might be influenced by personality or social experience. In order to examine this hypothesis, individual experience and any morphological

traits associated with outcome would need to be carefully controlled. This is because outcome is determined by the weaker individual. From my reading of the text this does not appear to have been the case. A more appropriate candidate would be to assess aggressiveness in relation to social experience and personality.

Why would you expect personality to be associated with morphological traits? Although you test the relationship with aggression (outcome?) I couldn't quite get the link with comb size etc. I realise you mention an association between slow changing 'states' and size in the Introduction but it wasn't clear where social experience entered into this (or if it did) and how this might influence personality or morphology. Can you comment and redraft the argument?

The numbers of individuals and numbers of personality tests given are very low. In general, a higher number of both participants and replicates are required to determine between individual differences and I think this needs to be acknowledged in the paper. I realise that the nature of the study requires that numbers be as low as possible from an ethical perspective but it does not address the question of replicates.

Did the authors investigate the effect of winning and losing on their results. I know these can be variable both as effects and across species. What is the current state of knowledge in fowl and how might this have impacted on their results?

I think the ideas tested in this paper are interesting, however, my reading of the paper left me somewhat confused as to the predictions being tested, and raised a number of questions that the authors should address before the paper is accepted.

Review form: Reviewer 3

Is the manuscript scientifically sound in its present form?

Yes

Are the interpretations and conclusions justified by the results?

Yes

Is the language acceptable?

Yes

Do you have any ethical concerns with this paper?

No

Have you any concerns about statistical analyses in this paper?

No

Recommendation?

Accept with minor revision (please list in comments)

Comments to the Author(s)

Numerous studies have explored cost and benefits of dominance, and documented the process of hierarchy formation. However, little is known regarding the proximate causes of social dominance. What makes an individual dominant over another, or to what extent social experience or intrinsic characteristics determine the ability to dominate, are questions that are highly relevant to understand social dynamics and to improve the management of captive

populations, for example. This paper explores the impact of social experience during development, and other intrinsic individual characteristics with relatively simple experiments. The paper is very interesting, it is nicely written and address the above mentioned questions specifically. Nevertheless, clarity of the manuscript, especially regarding experimental design could be much improved. It is not totally clear how the duels were run (L 55), 'each male', referring to intermediate birds of the grouped males, dominants from the pair, singly housed? Or all of them? paired for the duel with birds from the 'main population' which is not defined previously? What was the social status of the birds from the main population? It is really hard to follow when in fact the experimental design is not that complex. Looking at the results section, it is clear that the duels were Single vs Grouped, but then I do not understand what the role of the birds from the main population was (or was call main population the large groups, but seems improbable because of the mentioned bird age.. (L 55)?). Besides clarifying the text, I would suggest removing figure 1 which does not add much to the clarity of the experimental design and replace it by a table or figure that is self-explanatory regarding the experimental design and Id of the birds participating in the duels. One more consideration is that as explained by the authors, normally males dominate over females. However sometimes females dominate over males, therefore, the assumption that singly paired birds are dominants, might not be always correct. Did the authors prove that indeed was the case in this study? Are there any data supporting the dominance of singly housed males?

L 37 Unclear the assignment of females; To which groups, all of them? Please clarify.

L 54 The 6 and 7 males were they the intermediate of the grouped housed? or alternatively; subordinates and intermediate were pair with 6 or 7 males from main population depending on the year? If so, was it known the dominance status of the birds from the main population?

L 58 Please define clearly what are the opponent and focal birds.

Pp 4- L 30 Comb size, why was it estimated over length and not length and height?

Discussion

The discussion section addresses all main results but I found it a little superficial. Given the abundant literature on the matter I think authors can dig a little more on the implications of their results.

L 30-31 The authors do not mention the role of comb size as a status badge. Very dominant males do not need to incur in aggressive interactions as the dominance signalling should be sufficient normally to deter opponents of engaging in costly interactions that are likely to lose. Literature indicated that intermediate range birds will be the ones escalating in aggressive interactions most often.

L 50-51 It is extended from aggressive interaction data that social integration was limited in singly reared males. I do not think the authors have support for such statement. It gives the impression that sociality was only occurring between grouped males without considering the social relationships that singly housed males may have developed with the females within their own groups.

Tables 1 and 2 clearly shows differences between batches from 2014 and 2013 but I found not a single line trying to explain the reason for this difference.

Decision letter (RSOS-201213.R0)

Dear Dr Leimar

On behalf of the Editors, we are pleased to inform you that your Manuscript RSOS-201213 "Effects of social experience, aggressiveness and comb size on contest success in male domestic fowl" has been accepted for publication in Royal Society Open Science subject to minor revision in accordance with the referees' reports. Please find the referees' comments along with any feedback from the Editors below my signature.

Please submit your revised manuscript and required files (see below) no later than 7 days from today's (ie 20-Oct-2020) date. Note: the ScholarOne system will 'lock' if submission of the revision is attempted 7 or more days after the deadline. If you do not think you will be able to meet this deadline please contact the editorial office immediately.

on behalf of Dr Claudia Wascher (Associate Editor) and Kevin Padian (Subject Editor)

Associate Editor Comments to Author (Dr Claudia Wascher):

The presented study investigates the effect of social experience, aggressiveness, and morphological traits for competitiveness in social interactions in male domestic fowl. The reviewers find the experiment well designed and interesting, however they have some suggestions which hopefully will improve the clarity of the manuscript. I am looking forward to receive the revision.

Reviewer comments to Author:

Reviewer: 1

Comments to the Author(s)

The paper is not bringing breakthrough information, however, the labour and time intensive experiments are well designed and results are presented in clear and concise way. In addition to Fig. 1, an infographics describing experimental setup would be useful. Not applicable for this manuscript, however, it would be beneficial to have information about the testosterone levels in

tested males. Along with comb length, perhaps spur length and wattle size could add relevant information.

Reviewer: 2

Comments to the Author(s)

The present manuscript reports the results of an experiment that investigated the relationship between social experience and contest outcome, and secondarily, whether social experience influences personality according to the Introduction (the Discussion takes a different tack and states that the purpose of the study was to investigate the influence of social experience and personality on fighting ability). I think some serious re-drafting of the text is required as I had some difficulty understanding what variables were supposed to be influencing what. There is also the lack of consistency throughout the paper; for example, fight outcome and individual aggressiveness (which are not the same thing) are referenced but I was under the impression that outcome was the variable of interest.

I think the main problem I had was with the somewhat circular nature of the rationale underlying the experiment. It wasn't clear to me whether social experience was expected to influence personality. From my reading of the Introduction I think this is the case; however, we need to know the personality of the individual before they are exposed to the social group element of the procedure. This is not what has happened here where social groups were set up and personality subsequently tested.

I was interested in why the authors felt that outcome might be influenced by personality or social experience. In order to examine this hypothesis, individual experience and any morphological traits associated with outcome would need to be carefully controlled. This is because outcome is determined by the weaker individual. From my reading of the text this does not appear to have been the case. A more appropriate candidate would be to assess aggressiveness in relation to social experience and personality.

Why would you expect personality to be associated with morphological traits? Although you test the relationship with aggression (outcome?) I couldn't quite get the link with comb size etc. I realise you mention an association between slow changing 'states' and size in the Introduction but it wasn't clear where social experience entered into this (or if it did) and how this might influence personality or morphology. Can you comment and redraft the argument?

The numbers of individuals and numbers of personality tests given are very low. In general, a higher number of both participants and replicates are required to determine between individual differences and I think this needs to be acknowledged in the paper. I realise that the nature of the study requires that numbers be as low as possible from an ethical perspective but it does not address the question of replicates.

Did the authors investigate the effect of winning and losing on their results. I know these can be variable both as effects and across species. What is the current state of knowledge in fowl and how might this have impacted on their results?

I think the ideas tested in this paper are interesting, however, my reading of the paper left me somewhat confused as to the predictions being tested, and raised a number of questions that the authors should address before the paper is accepted.

Reviewer: 3

Comments to the Author(s)

Numerous studies have explored cost and benefits of dominance, and documented the process of hierarchy formation. However, little is known regarding the proximate causes of social

dominance. What makes an individual dominant over another, or to what extent social experience or intrinsic characteristics determine the ability to dominate, are questions that are highly relevant to understand social dynamics and to improve the management of captive populations, for example. This paper explores the impact of social experience during development, and other intrinsic individual characteristics with relatively simple experiments.

The paper is very interesting, it is nicely written and address the above mentioned questions specifically. Nevertheless, clarity of the manuscript, especially regarding experimental design could be much improved. It is not totally clear how the duels were run (L 55), 'each male', referring to intermediate birds of the grouped males, dominants from the pair, singly housed? Or all of them? paired for the duel with birds from the 'main population' which is not defined previously? What was the social status of the birds from the main population? It is really hard to follow when in fact the experimental design is not that complex. Looking at the results section, it is clear that the duels were Single vs Grouped, but then I do not understand what the role of the birds from the main population was (or was call main population the large groups, but seems improbable because of the mentioned bird age.. (L 55)?). Besides clarifying the text, I would suggest removing figure 1 which does not add much to the clarity of the experimental design and replace it by a table or figure that is self-explanatory regarding the experimental design and Id of the birds participating in the duels. One more consideration is that as explained by the authors, normally males dominate over females. However sometimes females dominate over males, therefore, the assumption that singly paired birds are dominants, might not be always correct. Did the authors prove that indeed was the case in this study? Are there any data supporting the dominance of singly housed males?

L 37 Unclear the assignment of females; To which groups, all of them? Please clarify.

L 54 The 6 and 7 males were they the intermediate of the grouped housed? or alternatively; subordinates and intermediate were pair with 6 or 7 males from main population depending on the year? If so, was it known the dominance status of the birds from the main population?

L 58 Please define clearly what are the opponent and focal birds.

Pp 4- L 30 Comb size, why was it estimated over length and not length and height?

Discussion

The discussion section addresses all main results but I found it a little superficial. Given the abundant literature on the matter I think authors can dig a little more on the implications of their results.

L 30-31 The authors do not mention the role of comb size as a status badge. Very dominant males do not need to incur in aggressive interactions as the dominance signalling should be sufficient normally to deter opponents of engaging in costly interactions that are likely to lose. Literature indicated that intermediate range birds will be the ones escalating in aggressive interactions most often.

L 50-51 It is extended from aggressive interaction data that social integration was limited in singly reared males. I do not think the authors have support for such statement. It gives the impression that sociality was only occurring between grouped males without considering the social relationships that singly housed males may have developed with the females within their own groups.

Tables 1 and 2 clearly shows differences between batches from 2014 and 2013 but I found not a single line trying to explain the reason for this difference.

===PREPARING YOUR MANUSCRIPT===

===PREPARING YOUR REVISION IN SCHOLARONE===

-- Ensure that your data access statement meets the requirements at <https://royalsociety.org/journals/authors/author-guidelines/#data>. You should ensure that you cite the dataset in your reference list. If you have deposited data etc in the Dryad repository, please only include the 'For publication' link at this stage. You should remove the 'For review' link.

-- If you have uploaded ESM files, please ensure you follow the guidance at <https://royalsociety.org/journals/authors/author-guidelines/#supplementary-material> to include a suitable title and informative caption. An example of appropriate titling and captioning may be found at https://figshare.com/articles/Table_S2_from_Is_there_a_trade-off_between_peak_performance_and_performance_breadth_across_temperatures_for_aerobic_sc_ope_in_teleost_fishes_/3843624.

Author's Response to Decision Letter for (RSOS-201213.R0)

See Appendix A.

RSOS-201213.R1 (Revision)

Review form: Reviewer 2

Is the manuscript scientifically sound in its present form?

Yes

Are the interpretations and conclusions justified by the results?

Yes

Is the language acceptable?

Yes

Do you have any ethical concerns with this paper?

No

Have you any concerns about statistical analyses in this paper?

No

Recommendation?

Accept as is

Comments to the Author(s)

The authors have done a fine job of addressing the comments I made on an earlier version of their ms. I have no further concerns.

Review form: Reviewer 3

Is the manuscript scientifically sound in its present form?

Yes

Are the interpretations and conclusions justified by the results?

Yes

Is the language acceptable?

Yes

Do you have any ethical concerns with this paper?

No

Have you any concerns about statistical analyses in this paper?

No

Recommendation?

Accept as is

Comments to the Author(s)

The clarity of the manuscript has been much improved and in my opinion it is suitable for publication

Decision letter (RSOS-201213.R1)

Dear Dr Leimar,

It is a pleasure to accept your manuscript entitled "Effects of social experience, aggressiveness and comb size on contest success in male domestic fowl" in its current form for publication in Royal Society Open Science. The comments of the reviewer(s) who reviewed your manuscript are included at the foot of this letter.

on behalf of Dr Claudia Wascher (Associate Editor) and Kevin Padian (Subject Editor)
openscience@royalsociety.org

Reviewer comments to Author:

Reviewer: 2

Comments to the Author(s)

The authors have done a fine job of addressing the comments I made on an earlier version of their ms. I have no further concerns.

Reviewer: 3

Comments to the Author(s)

The clarity of the manuscript has been much improved and in my oppinion it is suitable for publication

Dear Editor,

Thank you for your handling of our manuscript and for the many helpful suggestions for changes, which we feel have greatly helped us to improve our presentation. Our responses to the comments appear below, in blue and are interleaved with the comments by the associate editor and reviewers.

Best regards,

Olof Leimar and co-authors

Associate Editor Comments to Author (Dr Claudia Wascher):

The presented study investigates the effect of social experience, aggressiveness, and morphological traits for competitiveness in social interactions in male domestic fowl. The reviewers find the experiment well designed and interesting, however they have some suggestions which hopefully will improve the clarity of the manuscript. I am looking forward to receive the revision.

Thank you. We have taken all comments into consideration, which resulted in a number of changes to the manuscript, as detailed below and as seen in the track-changes version of the revised manuscript.

Reviewer comments to Author:

Reviewer: 1

Comments to the Author(s)

The paper is not bringing breakthrough information, however, the labour and time intensive experiments are well designed and results are presented in clear and concise way. In addition to Fig. 1, an infographics describing experimental setup would be useful. Not applicable for this manuscript, however, it would be beneficial to have information about the testosterone levels in tested males. Along with comb length, perhaps spur length and wattle size could add relevant information.

Thank you for pointing out that an infographics describing the setup is needed. In the revised version, we have included this as Figure 1, changing the numbering of the other figures accordingly. Hopefully, this will clear up some of the questions and comments provided by the other reviewers.

Regrettably, we do not have data on testosterone levels or on other morphological measurements, such as spur length.

Reviewer: 2

Comments to the Author(s)

The present manuscript reports the results of an experiment that investigated the relationship between social experience and contest outcome, and secondarily, whether social experience influences personality according to the Introduction (the Discussion takes a different tack and states that the purpose of the study was to investigate the influence of social experience and personality on fighting ability). I think some serious re-drafting of the text is required as I had some difficulty understanding what variables were supposed to be influencing what. There is also the lack of consistency throughout the paper; for example, fight outcome and individual aggressiveness (which are not the same thing) are referenced but I was under the impression that outcome was the variable of interest.

Thank you for pointing out that we need to clarify our presentation. It is correct that the contest outcome (contest success) is the main variable we are investigating. Aggressiveness, measured in assays before and after duels, is a variable that potentially correlates with contest success. This is alluded to in the title and in the second sentence of the abstract, but the reviewer rightly notes that in other places we have not been clear. To avoid confusion, we have made a number of changes. First, we modified our terminology, using the term contest success consistently. Second, we changed the final paragraph of the introduction, to better represent the aims of the study. Third, we made changes in the first paragraph of the discussion section to further clarify the aim of the study and what we conclude from it. Fourth, we made many small changes to make clear that we only claim that we found aggressiveness to be correlated with contest success; we do not claim to have demonstrated that aggressiveness is directly causing contest success.

In summary, in the comparison of contest success against opponent males, between Single and Group social treatments, the treatments were experimentally imposed, allowing conclusions to be drawn about causal relations, whereas the covariation we found of aggression and comb size with contest success is a correlation, for which we have not established causation. Further, concerning the comparison of contest success against opponent males between dominant and subordinate social experience, we point out in the second paragraph of the discussion section that these social experiences were not experimentally imposed. We hope that the changes we have made to the manuscript clarifies which relationships we are investigating and how we interpret our results.

I think the main problem I had was with the somewhat circular nature of the rationale underlying the experiment. It wasn't clear to me whether social experience was expected to influence personality. From my reading of the Introduction I think this is the case; however, we need to know the personality of the individual before they are exposed to the social group element of the procedure. This is not what has happened here where social groups were set up and personality subsequently tested.

It seems that we did not manage to make clear the role of the personality assays in our study in the previous version, and we have tried to improve this aspect of the presentation. As explained in our response above, our main aim is to investigate experimentally if social

treatment influences contest success. To achieve this aim, we do not need to measure the personality of males before they are exposed to the social treatment. However, as is mentioned in the final paragraph of the introduction section, it is conceivable that the social treatment influences contest success at least partly through changes in aggressiveness. For this reason it is of interest to examine if aggressiveness differs between social treatments, and for such an experimental comparison it is not necessary to measure aggressiveness of males before they are exposed to the social treatment.

Further, as we point out in the introduction section, from previous studies it is known that aggressiveness can covary with social dominance, and thus potentially with contest success, and this is the reason it is included in our analysis in Table 1 (and in Table 2). The other personality traits, measured in the novel arena assay, were included in the study to investigate if they correlated with aggressiveness (Table 3).

I was interested in why the authors felt that outcome might be influenced by personality or social experience. In order to examine this hypothesis, individual experience and any morphological traits associated with outcome would need to be carefully controlled. This is because outcome is determined by the weaker individual. From my reading of the text this does not appear to have been the case. A more appropriate candidate would be to assess aggressiveness in relation to social experience and personality.

Our reasons for investigating the hypothesis that contest success (contest outcome) could be influenced by social experience, and be related to aggressiveness, is described in the two last paragraphs of the introduction section, where we have made changes to improve the clarity of the presentation. As we mentioned in our response above, we believe that it is valid to compare contest success, as well as aggressiveness, between the Single and Group social treatments. Also, we measured traits such as aggressiveness, comb size and weight, and introduced these into the statistical analysis (e.g., Figure 1, Table 1).

Why would you expect personality to be associated with morphological traits? Although you test the relationship with aggression (outcome?) I couldn't quite get the link with comb size etc. I realise you mention an association between slow changing 'states' and size in the Introduction but it wasn't clear where social experience entered into this (or if it did) and how this might influence personality or morphology. Can you comment and redraft the argument?

It is not our aim to investigate if personality is related to morphology. We do investigate if social experience influences aggressiveness (as reported in subsection 4.2 of the results, where we found no effect). As mentioned above, our reason for thinking that the latter is of interest is that social experience might influence contest success at least partly through an effect on aggressiveness. We realise that our presentation lacked clarity in several places, and we hope that with our changes the main points of our study come across in a clearer way.

The numbers of individuals and numbers of personality tests given are very low. In general, a higher number of both participants and replicates are required to determine between individual differences and I think this needs to be acknowledged in the paper. I realise that

the nature of the study requires that numbers be as low as possible from an ethical perspective but it does not address the question of replicates.

The reviewer is correct that the number of focal males in our experiment was rather low. It would of course have been nice if a larger sample size could have been used, but the amount of work and time needed for each focal male was simply too great to achieve this. Nevertheless, with our relatively low number of focal males, we still found clear effects of the Single vs. Group treatment, as well as of aggressiveness and comb size. Perhaps a larger sample size would have resolved a possible difference between Subordinate and Dominant focal Pair males (Table 2). To make this possibility clear, we wrote the following in the second paragraph of the discussion: "However, because of a small sample size and perhaps random variation in contest outcomes it is not possible to draw any firm conclusions. It thus remains to be determined whether or not winning as compared to losing experiences during ontogeny in male domestic fowl has a longer-term influence on contest success." It seems to us that this gives a reasonable description of the current situation.

Concerning the number of personality tests, two replicate measures of a personality trait is sufficient to estimate consistency of behaviour. However, we have not included any analysis of this in the manuscript; we only show results on consistency of contest success (subsection 4.3, Figure 4).

Did the authors investigate the effect of winning and losing on their results. I know these can be variable both as effects and across species. What is the current state of knowledge in fowl and how might this have impacted on their results?

There is of course work on social hierarchy formation in domestic fowl, including such things as winner and loser effects. Although the fascinating questions of how winner and loser effects operate is not the specific focus of our study, we now cite work on fowl sociality in general (McDonald et al. 2017 and Pizzari and McDonald 2019). The latter of these includes a discussion of winner and loser effects in fowl.

Concerning an impact of winner and loser effects on our results, one might expect that our comparison of contest success between dominants and subordinates of Pair males would show an influence of winner and loser effects. However, we did not find any statistically significant effect (Table 2).

I think the ideas tested in this paper are interesting, however, my reading of the paper left me somewhat confused as to the predictions being tested, and raised a number of questions that the authors should address before the paper is accepted.

We hope that our changes have managed to clarify these issues.

Reviewer: 3

Comments to the Author(s)

Numerous studies have explored cost and benefits of dominance, and documented the process of hierarchy formation. However, little is known regarding the proximate causes of

social dominance. What makes an individual dominant over another, or to what extent social experience or intrinsic characteristics determine the ability to dominate, are questions that are highly relevant to understand social dynamics and to improve the management of captive populations, for example. This paper explores the impact of social experience during development, and other intrinsic individual characteristics with relatively simple experiments.

The paper is very interesting, it is nicely written and address the above mentioned questions specifically.

Thank you for these encouraging comments.

Nevertheless, clarity of the manuscript, especially regarding experimental design could be much improved. It is not totally clear how the duels were run (L 55), 'each male', referring to intermediate birds of the grouped males, dominants from the pair, singly housed? Or all of them? paired for the duel with birds from the 'main population' which is not defined previously?

Thank you for pointing out that we need to improve the clarity of our presentation. These experimental males are referred to as focal males in the manuscript. We have now checked our manuscript, making many small changes, ensuring that we are using the terminology focal males in a consistent way. We are sorry for any confusion caused by our terminology in the previous version.

What was the social status of the birds from the main population?

The social status of males from the main population varied. The second sentence of subsection 3.3 (Duels) now reads: "We implemented a design where a focal male met 6 (in the first round of duels) and 7 (in the second round of duels) unfamiliar randomly chosen conspecific opponent males from our main population (age 3-5 years)". Thus, the males from the main population were randomly selected and thus varied in their social status. We believe that this is a reasonable way of investigating the contest success of the focal males. We refer to the males from the main population as opponent males throughout the manuscript.

It is really hard to follow when in fact the experimental design is not that complex. Looking at the results section, it is clear that the duels were Single vs Grouped, but then I do not understand what the role of the birds from the main population was (or was call main population the large groups, but seems improbable because of the mentioned bird age.. (L 55)?). Besides clarifying the text, I would suggest removing figure 1 which does not add much to the clarity of the experimental design and replace it by a table or figure that is self-explanatory regarding the experimental design and Id of the birds participating in the duels.

We agree and hope that our added infographics (current Figure 1) and other changes, such as making sure that we refer to the social treatments, focal males and opponent males in a consistent way, helps to clarify these issues. The duels (contests) were between focal and opponent males. We made changes to Figure and Table captions to make this clear.

One more consideration is that as explained by the authors, normally males dominate over females. However sometimes females dominate over males, therefore, the assumption that singly paired birds are dominants, might not be always correct. Did the authors prove that indeed was the case in this study? Are there any data supporting the dominance of singly housed males?

We now note, at the end of the second paragraph of subsection 3.2, that males dominated over females in the social treatments. The reviewer is correct that females can dominate over males in domestic fowl, but this is rare and in our experience, both with regard to our breed of domestic fowl and red junglefowl, does not happen with healthy, sexually mature males.

L 37 Unclear the assignment of females; To which groups, all of them? Please clarify.

Hopefully, our new Figure 1 and our changes to the first paragraph of subsection 3.2 make this clear.

L 54 The 6 and 7 males were they the intermediate of the grouped housed? or alternatively; subordinates and intermediate were pair with 6 or 7 males from main population depending on the year? If so, was it known the dominance status of the birds from the main population?

These were males from the main population. We are sorry for the confusion and we hope our changes will clear it up. The 6 or 7 males from the main population mentioned in subsection 3.3 are now consistently referred to as opponent males. As stated in that subsection, they were randomly selected from the main population each year and were kept in a separate flock. Dominance positions were not scored in this flock.

L 58 Please define clearly what are the opponent and focal birds.

As mentioned above, we hope our changes clarifies this issue.

Pp 4- L 30 Comb size, why was it estimated over length and not length and height?

Given the morphology of domestic fowl combs, it is easier to get an accurate measurement of the length than of height. This measurement has also been used previously, and been shown to relate to social status, which is why we decided to use it in the current study.

Discussion

The discussion section addresses all main results but I found it a little superficial. Given the abundant literature on the matter I think authors can dig a little more on the implications of their results.

It is not clear to us from this comment precisely what the reviewer is after, but we added a citation to McDonald et al. (2017) in the introduction and to Pizzari and McDonald (2019) in the discussion. The latter contains a broad discussion of social interactions in red junglefowl and domestic fowl, including such things as the importance of aggressiveness.

L 30-31 The authors do not mention the role of comb size as a status badge. Very dominant males do not need to incur in aggressive interactions as the dominance signalling should be sufficient normally to deter opponents of engaging in costly interactions that are likely to lose. Literature indicated that intermediate range birds will be the ones escalating in aggressive interactions most often.

This is an interesting point. However, our previous work shows that, during the initial establishment of social status, males that become top ranked need to show aggressive behaviour to reach this position. Concerning comb size as a status signal, we briefly refer to this possibility in the final sentence of the third paragraph of the discussion section. Note, however, that our study finds that aggressiveness varies seemingly independently of comb size and is positively related to contest success. In order to get a high score in our aggression assay, a male needs to show aggressive behaviour, and this argues against the idea of a badge of status that is sufficient to determine the dominance position.

L 50-51 It is extended from aggressive interaction data that social integration was limited in singly reared males. I do not think the authors have support for such statement. It gives the impression that sociality was only occurring between grouped males without considering the social relationships that singly housed males may have developed with the females within their own groups.

We now state more clearly that males in the Single treatment had limited experience specifically of interactions with other males. Beyond that we do not make any claims about social integration for the focal males of our study.

Tables 1 and 2 clearly shows differences between batches from 2014 and 2013 but I found not a single line trying to explain the reason for this difference.

We do not know the reason for this difference, which we now state at the end of subsection 4.1.